**www.cambridge.org/ext**

Community ecology; ecosystem; evolution; macroevolution; volcanic eruptions

**Corresponding author:**
S.A.F. Darroch;
Email: simon.a.darroch@vanderbilt.edu

# Causes and consequences of end-Ediacaran extinction: An update

Simon A.F. Darroch[1,2,3] ⓘ, Emily F. Smith[4], Lyle L. Nelson[5], Matthew Craffey[6], James D. Schiffbauer[7,8] ⓘ and Marc Laflamme[9]

[1]Department of Earth and Environmental Sciences, Vanderbilt University, Nashville, TN, USA; [2]Evolutionary Studies Institute, Vanderbilt University, Nashville, TN, USA; [3]Senckenberg Research Institute and Museum of Natural History, Frankfurt, Germany; [4]Department of Earth and Planetary Sciences, Johns Hopkins University, Baltimore, MD, USA; [5]Department of Earth Sciences, Carleton University, Ottawa, ON, Canada; [6]Department of Biological Sciences, University of Nebraska, Lincoln, NE, USA; [7]Department of Geological Sciences, University of Missouri, Columbia, MO, USA; [8]X-Ray Microanalysis Laboratory, University of Missouri, Columbia, MO, USA and [9]Department of Chemical and Physical Sciences, University of Toronto Mississauga, Mississauga, ON, Canada

## Abstract

Since the 1980s, the existence of one or more extinction events in the late Ediacaran has been the subject of debate. Discussion surrounding these events has intensified in the last decade, in concert with efforts to understand drivers of global change over the Ediacaran–Cambrian transition and the appearance of the more modern-looking Phanerozoic biosphere. In this paper we review the history of thought and work surrounding late Ediacaran extinctions, with a particular focus on the last 5 years of paleontological, geochemical, and geochronological research. We consider the extent to which key questions have been answered, and pose new questions which will help to characterize drivers of environmental and biotic change. A key challenge for future work will be the calculation of extinction intensities that account for limited sampling, the duration of Ediacaran 'assemblage' zones, and the preponderance of taxa restricted to a single 'assemblage'; without these data, the extent to which Ediacaran bioevents represent genuine mass extinctions comparable to the 'Big 5' extinctions of the Phanerozoic remains to be rigorously tested. Lastly, we propose a revised model for drivers of late Ediacaran extinction pulses that builds off recent data and growing consensus within the field. This model is speculative, but does frame testable hypotheses that can be targeted in the next decade of work.

## Impact statement

The majority of extinction-based paleontological research over the last four decades has focused on the 'big 5' mass extinctions of the Phanerozoic. In parallel, however, geologists and paleontologists working in the Precambrian have mulled the existence of one or more pulses of extinction (and potentially 'mass extinction') in the latest Neoproterozoic (~574–539 million years ago) shortly before the onset of the Cambrian. These episodes of global biotic turnover removed the mysterious Ediacara biota, as well as groups of more recognizable animal fossils. In this review, we summarize the history of ideas and research surrounding these events, as well as recent work in a range of fields that is attempting to identify the drivers – both biotic and abiotic – of extinction. We outline four key questions which, we argue, will help us to compare the causes and consequences of Ediacaran extinction alongside the Phanerozoic 'Big 5', and which will help us decide whether the 'Big 5' might eventually become the 'Big 6' (or the 'Big 7', if the current biodiversity crisis is considered). Finally, we propose a model for drivers of late Ediacaran extinction that builds off recent data. This model is speculative, but frames testable hypotheses that will help determine the role these events may have played in the Ediacaran–Cambrian emergence of the modern-looking biosphere, and thus the extent to which Ediacaran extinction and the Cambrian explosion may be linked.

## Introduction

The Ediacaran–Cambrian (E–C) transition arguably marks the most important geobiological revolution of the past billion years, characterized by large perturbations to global geochemical cycles, a permanent step-change in the character of the sedimentary record, the rise of macroscopic eukaryotic life, and potentially one or more pulses of mass extinction. Although all aspects of this interval have been the subjects of intense research efforts over the last three decades, the existence of putative biotic turnover events in the latest Ediacaran has received particular attention. These events may not only have played a crucial role in fueling evolutionary radiation during the 'rise of animals' and acted as a powerful influence on the appearance of metazoan

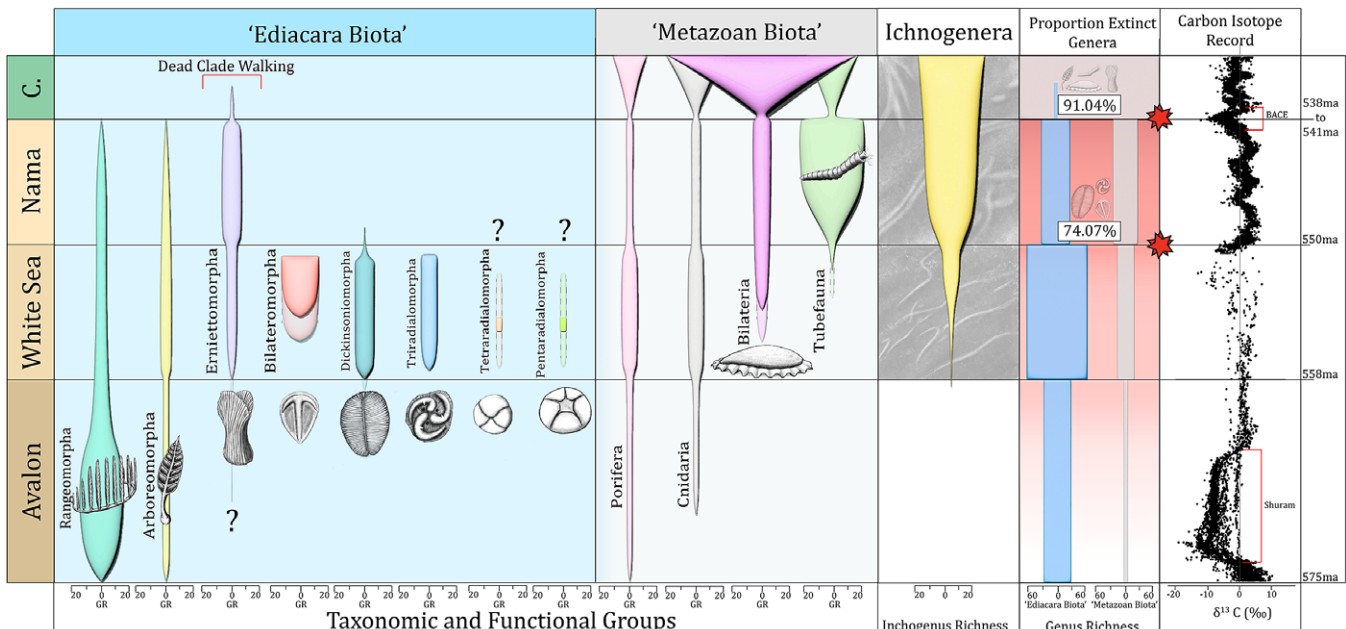

**Figure 1.** Updated summary figure illustrating the stratigraphic distribution and diversity among groups of Ediacara biota, as well as metazoans, bilaterian ichnogenera, and a $\delta^{13}$C curve (compiled from Yang et al. (2021), Bowyer et al. (2022), and references therein). The stratigraphic ranges of the Pentaradialomorpha and Tetraradialomorpha are currently uncertain, but currently constrained by a detrital zircon age of 556 ± 24 Ma obtained from the Bonney Sandstone in South Australia (Ireland et al., 1998). Solid colors represent minimum age estimates (where available), while shaded regions represent uncertain range estimates where taxa are found beneath (or between) dated horizons. Extinction intensities – as percentage of genera lost – are given for the two putative extinction pulses at the White Sea-Nama and the E–C boundaries; intensities were calculated by simply measuring the proportion of surviving genera over total genera in the preceding assemblage zone (although see discussion in the text surrounding problems with calculating these transition).

ecosystems with a more modern-looking structure (Knoll and Carroll, 1999; Droser et al., 2017; Darroch et al., 2018a,b), but they also have invited comparison with the 'Big 5' mass extinctions of the Phanerozoic (Raup and Sepkoski, 1982), and thus may yield more general lessons about the causes and consequences of these catastrophic events in Earth's history.

Five years ago, Darroch et al., 2018a, summarized the evidence for one or more pulses of extinction in the late Ediacaran and presented a series of key questions that would be crucial for driving knowledge forward in this field. Since then many of these questions have been explored, with new paleontological, geochemical, and geochronological datasets providing the scaffolding required for building our understanding of this interval. In this review we summarize recent work surrounding the end-Ediacaran extinction events, examine the extent to which the questions posed in 2018 have been answered, and propose new questions, challenges, and research avenues that will continue to illuminate the changes that occurred over the E–C transition.

## Ediacaran fossils and early animals

The late Ediacaran is characterized by the presence of macroscopic body fossils that are typically categorized as belonging to one of two faunas: either (1) 'Ediacara biota' – an enigmatic collection of soft-bodied organisms with uncertain relationships to extant animal phyla, and which have been subdivided into morphogroups (see, e.g., Erwin et al., 2011; Laflamme et al., 2013); or (2) true metazoans – referring to fossils that can be more readily allied with living animal groups. However, this subdivision within Ediacaran organisms is becoming increasingly more obsolete with recent developmental (Gold et al., 2015), and phylogenetic (Dunn et al., 2021) data

suggesting that many representatives of the Ediacara biota are likely stem-group members of known eumetazoan clades. On the other hand, given that these two different categories of organisms appear and disappear at different times in the fossil record (see, e.g., Figure 1) and possess strong morphological differences (including, for example, the presence/absence of a body plan that is present among extant phyla), classifying Ediacaran-aged taxa as 'Ediacara biota' vs. 'metazoans' is arguably still useful, and provides a heuristic model with which to explore their faunal dynamics. So, while we refer to 'Ediacara biota' and metazoans over the course of this review, we emphasize that this does not preclude members of the Ediacara biota as belonging to animal clades.

## History

The history of thought surrounding the existence of end-Ediacaran extinction events is closely linked to work defining, dating, and characterizing the base of the Cambrian. Early attempts to define this boundary were spearheaded by decades of dedicated stratigraphic, paleontological, and geochronological studies conducted by the International Geoscience Programme (IGCP), reviewed nearly 30 years ago by Brasier et al. (1994). Nevertheless, these efforts were overwhelmingly focused on the subsequent 'explosion' of animal phyla in the early Cambrian, rather than the disappearance of Ediacaran soft-bodied organisms below the boundary. This is perhaps not surprising given that, prior to propositions by Seilacher (1984, 1985, 1989, 1992), the Ediacara biota were overwhelmingly interpreted as belonging to extant metazoan groups (e.g., Glaessner, 1984; Gehling, 1991). As such, the fossil record of the E–C transition could be satisfactorily explained as the result of taphonomic biases towards the preservation of biomineral shells,

teeth, and bones (Gehling, 1999). In contrast, Seilacher (1989) was largely alone in arguing that many, if not all, Ediacaran fossils represented neither true metazoans nor their earliest stem ancestors, and therefore must have suffered an extinction at some point during the E–C transition.

The case for extinction was reinvigorated by chemostratigraphic studies that identified multiple, large carbon isotope fluctuations in the late Neoproterozoic potentially tied to major upheavals in the carbon cycle; one that particularly stands-out as potentially coeval with extinction is a negative excursion that reaches values as low as −9 ‰, known now as the basal Cambrian carbon isotope excursion (or 'BACE'), which coincides with the E–C boundary (Kirschvink et al., 1991; Knoll and Walter, 1992; Narbonne et al., 1994). In the decade that followed the "Decision on the Precambrian-Cambrian boundary stratotype" (Brasier et al., 1994), notable discussion on a transitional Ediacaran–Cambrian extinction arose. For instance, noting parallels with the Permo-Triassic boundary, Knoll and Carroll (1999) stated a clear case for a mass extinction separating the Ediacaran and Cambrian faunas – a case that was only strengthened with the recognition that the BACE event also coincided with the global and synchronous disappearance of biomineralizing fossils that characterize the latest Ediacaran (Amthor et al., 2003). The presence of earlier extinction events, however, only became recognized with more focused biostratigraphic work, and, in particular, attempts to stratigraphically subdivide the late Ediacaran.

Waggoner (1999, 2003) identified three broad communities of Ediacara biota, which are still broadly thought to represent three chronologically and environmentally distinct assemblages. From oldest to youngest these are: (1) the Avalon Assemblage (~574–558 Ma), characterized by deep-water communities (Narbonne, 2005; Liu et al., 2015); (2) the White Sea Assemblage (~558–550 Ma), which represents the apex of diversity and disparity among Ediacara biota (Grazhdankin, 2004; Droser and Gehling, 2015); and (3) the Nama Assemblage (~550–538 Ma), which records a drop in the diversity of Ediacara biota, alongside an expansion in several modes of metazoan 'ecosystem engineering' including increased trace fossil diversity, the advent of macroscopic biomineralization, and widespread suspension feeding (Germs, 1972; Wood and Curtis, 2014; Schiffbauer et al., 2016; Darroch et al., 2018a,b). Due to the apparent loss in diversity among Ediacara biota, the transition from the White Sea assemblage to the Nama assemblage has also been suggested as recording an extinction event. Although discussion surrounding biotic turnover at the White Sea–Nama transition has intensified recently (e.g., Darroch et al., 2018a,b; Tarhan et al., 2018; Muscente et al., 2019; Evans et al., 2022), the formerly recognized 'Kotlinian crisis' in the southern Urals and East European Platform, which removed diversity among Ediacara biota prior to the appearance of biomineralizing metazoans like *Cloudina* (Brasier, 1992), may be time-equivalent with this transition (Grazhdankin, 2014).

The last 10 years have seen an abundance of work on putative late Ediacaran extinction events, focusing on the possible causes and consequences of extinction pulses, as well as to what extent extinction may be linked to the Cambrian explosion. Key work in this area is summarized below.

## Late Ediacaran bioevents and extinction models

Darroch et al. (2018a) argued for two pulses of Ediacaran extinction: one between the White Sea and Nama assemblages at ~550 Ma, and another at the E–C boundary itself. This inference was supported by Muscente et al. (2019), who used a network analysis of fossil communities together with their associated paleoenvironments to demonstrate that turnover was unlikely to be the result of a secular facies bias (see also Evans et al., 2022). The first extinction pulse apparently removed many of the most charismatic (and enigmatic) groups of Ediacara biota best known from White Sea-aged fossil localities in South Australia and Russia – principally the dickinsoniomorphs, triradialomorphs, tetraradialomorphs, and bilateromorphs – leaving a relatively species-poor assemblage dominated by erniettomorphs, arboreomorphs, and rangeomorphs in Nama-aged strata. Interestingly, genus richness among White Sea and Nama-aged rangeomorphs is also substantially lower than from the Avalon, suggesting that, while the rangeomorphs survived the first pulse of extinction, they were nonetheless negatively impacted. In contrast, genus richness among the erniettomorphs is equivalent or even potentially higher in the Nama, suggesting a positive response to the removal of White Sea-aged shallow-marine biocoenoses, and/or to the environmental conditions that pervaded the terminal Ediacaran. Although extinction is markedly focused within specific groups of Ediacara biota, many metazoan genera present in the White Sea were also affected (Figure 1). In concert with the disappearance of Ediacara biota, the first extinction pulse is also marked by the widespread appearance of more recognizable metazoans, including increased diversity (and/or behavioral disparity) in bilaterian tracemakers (Mángano and Buatois, 2014; Darroch et al., 2021), the appearance of tube-dwelling animals with debated affinities (see, e.g., Schiffbauer et al., 2020; Shore et al., 2020), calcifying and sessile lophotrochozoans (Shore et al., 2021), and rare body fossils of segmented bilaterians plausibly representing early annelids or panarthropods (Chen et al., 2019) (Figure 2). Unlike representatives of the Ediacara biota, these organisms can be more confidently allied with groups and lineages that persisted into the Cambrian (e.g., Yang et al., 2016, 2020). Given the apparent vermiform character of much of this Nama-aged metazoan fauna, whether preserved as body- or trace fossils, Schiffbauer et al. (2016) referred to this interval as 'Wormworld'.

An apparent second extinction pulse occurs at the E–C boundary, demarcated by the disappearance of almost all remaining Ediacara biota, as well as much of the metazoan fauna that characterizes the Nama Assemblage (in particular the calcifying taxa *Cloudina* and *Namacalathus* (Amthor et al., 2003; Smith et al., 2016, as well as cosmopolitan tube-dwelling forms such as *Shaanxilithes* and *Gaojiashania* (Zhu et al., 2017)). We note that estimates of extinction intensity – particularly with respect to tube-dwelling taxa – over this pulse is complicated by a lack of consensus in taxonomic studies, and thus to what extent latest Ediacaran and earliest Cambrian tubefauna may be related. For example, recent studies suggest that some Ediacaran-type biomineralizing taxa may persist into the early Cambrian (e.g., Zhu et al., 2017; Yang et al., 2021), albeit in limited localities and numbers (Cai et al., 2019). By way of contrast, few late Ediacaran taxa have unambiguously been identified from the Cambrian, and many that do are tentative descendants (e.g., cambroctoconids), and/or re-appear in considerably modified form (see, e.g., Park et al., 2021). In general, more focused systematic work on the affinities of, and relationships between, late Ediacaran and early Cambrian tubefauna is sorely needed (Schiffbauer et al., 2022).

In terms of what may have driven these two events, Darroch et al., 2018a, discussed evidence for two hypotheses – termed 'catastrophe' and 'biotic replacement' – representing the summation of ideas and data given in previous studies (principally Amthor et al., 2003; Erwin and Tweedt, 2012; Laflamme et al., 2013; and Darroch et al., 2015). The 'catastrophe' model suggested that

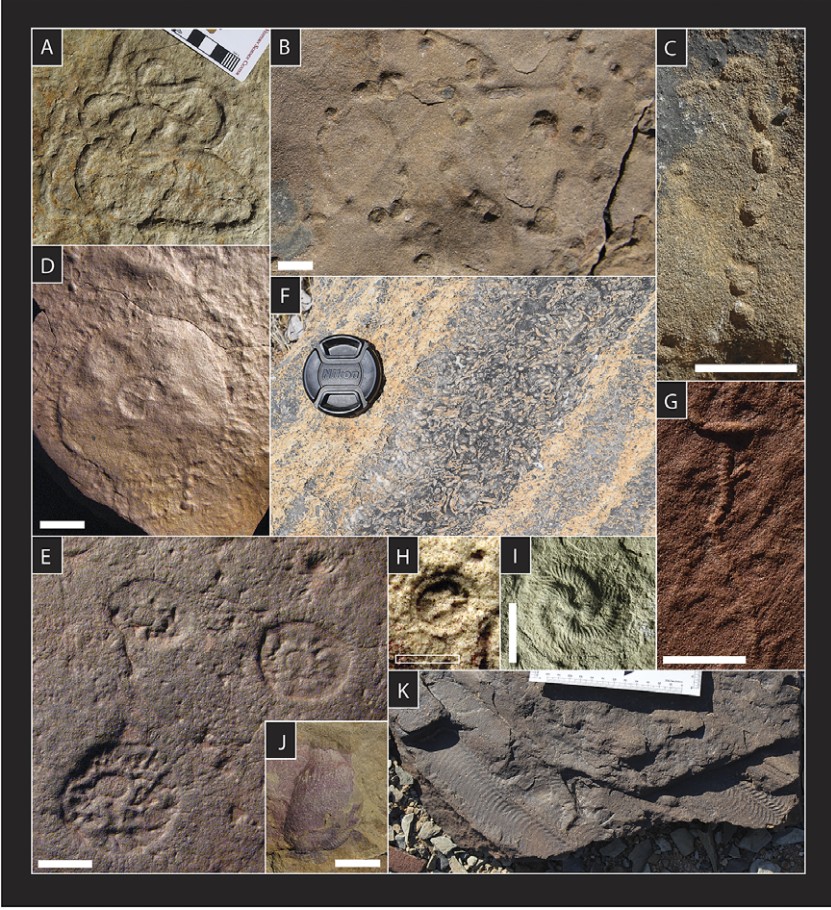

**Figure 2.** Putative late Ediacaran ecosystem engineers, including bilaterian tracemaking behaviors that involve sediment 'bulldozing' and biomixing (A - *Parapsammichnites*), bioirrigation (B-C - large treptichnids), and suspension feeders such as *Paleophragmodictya* (D-E; sp. nos. P32338 and P32332-P32352 respectively, South Australia Museum), biomineralizing *Cloudina* (F), and other unidentified tubefauna (G). Lastly, many Ediacara biota may have also had important ecosystem engineering impacts; the enigmatic taxa *Ernietta* (J), *Arkarua* (H; sp. no. P26768, South Australia Museum), *Tribrachidium* (I; sp. no. N3993/5056, Palaeontological Institute, Moscow) and *Pteridinium* (K) are all also thought to have functioned as suspension feeders, and thus played a crucial role in forging energetic links between the pelagic and benthic realms (Cracknell et al., 2021; Darroch et al., 2022). Specimens shown in A–C, F, G, J and K from the Nama Group of southern Namibia (all Urusis Fm., with the exception of *Ernietta* shown in J from the Dabis Fm.), and photographed in the field. Filled scale bars = 1 cm, open scale bars = 5 mm.

Ediacaran extinction events were driven by environmental perturbations, reflected in the negative carbon isotope excursions during the E–C transition. This model invokes parallels with several of the Phanerozoic 'Big 5' extinctions, in particular those coinciding with the Permian–Triassic and Triassic-Jurassic boundaries. In contrast, 'biotic replacement' suggested that the extinction events were instead driven by the emergence of new metazoan ecosystem engineering behaviors and their associated downstream geobiological impacts, which permanently altered marine environments in a fashion that was deleterious to soft-bodied Ediacara biota. There are fewer clear parallels for this process in the Phanerozoic; however, one hypothesized cause of the late Devonian mass extinction centers on the initial radiation of terrestrial forests, which significantly influenced weathering patterns leading to eutrophication, anoxia, and prolonged intervals of ecological stress (Algeo and Scheckler, 1998; Lu et al., 2021). This model, therefore, identifies the emergence of new ecosystem engineers as the ultimate driver of mass extinction, albeit through a complex series of terrestrial–marine teleconnections (e.g., Lu et al., 2021). Other studies have focused on the impact of humans as ecosystem engineers and as a driver of the on-going '6th mass extinction' (e.g., Yeakel et al., 2020; Pineda-Munoz et al., 2021), potentially raising interesting parallels between

the first and most recent mass extinctions of macroscopic life. In general, the extent to which the emergence of new ecosystem engineering behaviors in deep time had led to extinction, or instead evolutionary radiations, is a question that has long required more focused work (see, e.g., Erwin, 2008).

### Do these intervals of biotic turnover represent ('mass') extinctions at all?

Our understanding of the geochronology, chemostratigraphy, and biostratigraphy of the late Ediacaran has increased substantially in the last 10 years. However, given uncertainties surrounding stratigraphic correlation between sites (Xiao et al., 2016), the placement of key boundaries (Nelson et al., 2022), and the mechanisms of fossil preservation (Laflamme et al., 2013; Slagter et al., 2022; Gibson et al., 2023), a reasonable question is: do these apparent intervals of biotic turnover really represent extinction intervals (or more specifically, 'mass' extinctions) at all? The iconic Ediacaran fossil sites in South Australia illustrate some of these issues; they are among the best-studied Ediacaran localities, are frequently taken to epitomize the diversity and community structure of

'White Sea'-aged assemblages, and almost always make their way into analyses of Ediacaran diversity through time (see, e.g., Laflamme et al., 2013; Darroch et al., 2015; Darroch et al., 2018b; Eden et al., 2022; Evans et al., 2022). However, beyond post-dating the Shuram-Wonoka isotope excursion and a U–Pb detrital zircon age of 556 ± 24 Ma obtained from the underlying Bonney Sandstone (Ireland et al., 1998), the age of the principal fossiliferous horizons in the Ediacara Member are unconstrained. Given that South Australia preserves a number of Ediacaran morphogroups thought to be disappeared over the White Sea-Nama transition, this is perhaps not a trivial barrier to inferring an extinction event.

Mass extinctions are typically identified as episodes of anomalously high rates of taxonomic loss, occurring on global scales, that are approximately synchronous over a relatively short interval of geological time (exactly how 'short' is an evolving field, but currently thought to be ~$10^5$ years; see Burgess et al., 2014). We argue that the White Sea-Nama transition and E–C boundary currently satisfy one of these three criteria – specifically, being a global vs. regional signal. Older work (Laflamme et al., 2013; Boag et al., 2016) has suggested – and more recent work (Boddy et al., 2022; Evans et al., 2022) has confirmed – that there are no obvious geographical, facies, or preservational biases that can readily explain the loss of taxa, nor account for the observation that entire morphogroups (dickinsonimorpha, bilateromorpha, etc.) are lost, and thus extinction is apparently taxonomically clustered. The number and wide geographic spread of dated Nama-aged fossil sites also provides some support for dismissing the notion that White Sea-aged communities persist and remain widespread into the Nama. With respect to the issue with South Australian fossils mentioned above, it is thus far more likely that these communities are White Sea in age (coeval with well-dated horizons in Russia), rather than a totally unique Nama-aged locality.

The question as to rates (and magnitude) of taxonomic loss is harder to address. Calculating the simple proportion of surviving genera (over total genera in the preceding assemblage zone) gives genus extinction intensities of 74.1% for the White Sea-Nama (WS–NM) transition, and 91% over the Nama-Fortunian (NM–FN) – magnitudes noted by previous studies as being comparable to those estimated for many of the 'Big 5' (see, e.g., Evans et al., 2022). However, raw percentages are strongly biased by variations in sampling intensity, and quantifying extinction rates over the late Ediacaran is fraught with other difficulties stemming from the character of the Ediacaran fossil record, and ongoing difficulties with stratigraphic correlation and subdivision. For example, from our occurrence dataset (Figure 1; taxa and references provided in Supplementary Material) we can argue for major extinctions across the White Sea-Nama and E–C boundaries utilizing a simple proportion of extinct genera, but we do not have much confidence in Ediacaran per capita extinction rates using common methods such as Foote (1999), Alroy (2008), or Alroy (2014). The principal issue is that these methods utilize the proportion of taxa crossing boundaries for their extinction rate estimates in various patterns (e.g., boundary crossers, 3-timers, and gap fillers spanning 3 intervals with no detections in the middle) and the Ediacaran is dominated by taxa that occur in a single assemblage zone. This likely results from limited sampling, high turnover, and the long duration of these zones (~10–15 Ma). Thus, the effective sample size for an analysis of per capita Extinction rates across the Ediacaran is very small, leading to estimates of per capita extinction at the E–C boundary that have dubious reliability (for example: 2.48 per Foote, 1999, 3.04 per Alroy, 2014, utilizing the R package 'divDyn' Kocsis

et al., 2019). The most direct quantitative solution would be to examine extinction rates at finer timescales than assemblage zones, but there is – as yet – no unifying framework for subdividing these intervals, and there is limited chronostratigraphic data to applying such a framework across global collections. We can only argue, for now, for the presence of high extinction at the E–C transition based on apparent turnover patterns, but providing firm quantitative support will require additional work, and (potentially) the application of other methods for extinction rate modeling.

The last criterion – that extinction is rapid and synchronous – is, similarly hard to satisfy, and something discussed in more detail below (see the section titled 'Key questions in 2023').

Another recent challenge to the existence of late Ediacaran extinction events has come from phylogenetic modeling, framing biotic patterns over the E–C transition as an artifact of evolutionary patterns and stem- vs. crown-group diversity dynamics (i.e., suggesting that Ediacaran extinction and Cambrian explosion are different facets of the same process). For example, Budd and Mann (2020) have used birth-death models to argue that, if all Ediacara biota are viewed as stem-group members of extant bilaterian clades (and assuming a high level of background extinction), then the proportion of diversity within the total group can be quickly 'drowned' by the crown, thus mimicking a mass extinction. However, these models invoke a consistent rate of background extinction, and so cannot explain the synchronous and global loss of multiple morphogroups over, for example, the White Sea–Nama transition. Consequently, these models do not provide a good match for the observed diversity trends, although we note that Budd and Mann's (2020) models incorporating mass extinction events do provide a match, and so may be relevant for strikingly different reasons.

## Key questions in 2018

Darroch et al. (2018a) emphasized that the 'catastrophe' and 'biotic replacement' models were not mutually exclusive, but noted that each bring contrasting predictions, and moreover could be tested by addressing four key questions. Below, we briefly re-cap these questions (along with appropriate context), before reviewing recent work in these areas and assessing to what extent these questions have been answered.

*1. What, and when, was the Shuram?* In the Phanerozoic, several of the 'Big 5' mass extinctions are associated with perturbations to global geochemical cycles, which in turn are recorded in global isotope records (see, in particular, large igneous provinces as drivers of the Permian–Triassic (Shen et al., 2011) and Triassic–Jurassic (Ruhl et al., 2011) mass extinctions). Consequently, the existence of a large carbon isotope excursion in the late Neoproterozoic -– the Shuram event – has long been suspected as a potential source of environmental stress (see discussion in Tarhan et al., 2018). The Shuram is among the largest negative carbon isotope excursions in Earth history, with carbonate $\delta^{13}C$ values as low as −12‰ recorded on multiple paleocontinents (e.g., Grotzinger et al., 2011). It has been proposed that this excursion records massive perturbations to global geochemical cycles (e.g., Fike et al., 2006), which could plausibly represent a source of physiological stress and a possible driving mechanism for late Ediacaran extinction. However, in 2018 there were poor radiometric age constraints for the onset, duration, and recovery from the Shuram excursion, limiting efforts to test for temporal correlation with extinction. Furthermore, there was

disagreement over whether the excursion should be interpreted as a primary marine (e.g., Husson et al., 2015) or diagenetic (e.g., Knauth and Kennedy, 2009) signal.

*2. What was the BACE?* Similar to the Shuram, the BACE is a large negative carbon isotope excursion ($\delta^{13}$C values < −6‰) recorded in multiple localities worldwide. In 2018 the BACE had better age constraints than the Shuram and was recognized as coinciding with a second extinction pulse at the E–C boundary (Amthor et al., 2003). Like the Shuram, however, there were outstanding questions as to what the excursion represented, what its precise timing and duration was, and whether it was primary or diagenetic, locally or globally controlled. All of these uncertainties limited interpretations for causal linkages between excursion and extinction.

*3. Can we disentangle correlation* versus *causation in late Ediacaran extinction events?* This question builds from the previous two in emphasizing the need for plausible cause and effect in extinction studies – a standard set by workers over the last decade on the Permian–Triassic extinction involving integrated geochronology, geochemistry, and paleontology (e.g., Shen et al., 2011; Burgess et al., 2014; Clarkson et al., 2015). In 2018, although there was potential temporal correlation when discussing both the 'catastrophe' and 'biotic replacement' models, evidence for causation was lacking. For example, with the 'catastrophe' model there was little idea as to what E–C carbon isotope excursions represented, precluding discussion of links between environmental change and sources of biotic stress. Likewise, with 'biotic replacement' there were significant knowledge gaps surrounding how the Ediacara biota and emerging metazoan fauna interacted, both as individual taxa and within communities. Consequently, there was minimal evidence for biotic interactions – antagonistic or otherwise – and thus no substantiated mechanism for a biotic driver of extinction.

*4. What role did the end-Ediacaran extinction play in the Cambrian Explosion?* Several of the Phanerozoic 'Big 5' mass extinctions were followed by radiation in surviving clades, expansions of morphologic disparity, and rapid diversification of new taxa (e.g., post-K/Pg radiations of mammals (O'Leary et al., 2013), birds (Ksepka et al., 2017), and mollusks (Krug and Jablonski, 2012). An intriguing question, therefore, is to what extent the Cambrian explosion could have been triggered (or perhaps driven) by late Ediacaran extinction pulses as a response to an 'ecological vacuum' (e.g., Knoll and Carroll, 1999).

## Have these questions been answered?

Questions 1 and 2 ('what, and when, were the Shuram and BACE isotope excursions?') have arguably received the most attention over the last five years, with new geochemical and geochronological data bringing these events into sharper focus. With respect to the Shuram, recent Re-Os dates from Northwest Canada and Oman have demonstrated that: (1) on separate paleocontinents, the excursions are synchronous within the error of these radioisotopic measurements, and (2) the excursion lasted <6.7 ± 5.6 million years from c. 574 Ma to c. 567 Ma (Rooney et al., 2020). Additional radioisotopic and chemostratigraphic data from Newfoundland are consistent with this finding and suggest that the Shuram carbon isotope excursion began after 571 Ma and ended before 562 Ma (Canfield et al., 2020). These new data demonstrate that this perturbation did not coincide with the White Sea–Nama transition (at c. 550 Ma (Bowring et al., 2007), as previously suggested, and, furthermore, demonstrate that it postdated both the Gaskiers

glaciation and the earliest dated macrofossils of the Avalon assemblage (Macdonald et al., 2013; Pu et al., 2016; Matthews et al., 2021).

While the ultimate cause of the Shuram excursion remains contentious, many have suggested that mechanisms implicate changes in marine redox conditions (e.g., Fike et al., 2006; Zhang et al., 2019; Li et al., 2020) and/or changes in the locus of primary productivity (e.g., Busch et al., 2022). These changes are not mutually exclusive and could relate to external factors such as fluctuations in eustatic sea level (Busch et al., 2022) and/or nutrient availability (Cañadas et al., 2022). Other work has challenged the interpretation that large Neoproterozoic carbon isotope excursions record the global dissolved inorganic reservoir composition, but suggest they are still coeval responses to external forcings, such as primary production and sea level changes (e.g., Ahm et al., 2019). Regardless of its origin, at present it seems unlikely that the cause of the Shuram played any role in driving late Ediacaran extinction pulses (save, perhaps, for a decline in acanthomorphic acritarch assemblages (Ouyang et al., 2019; Yang et al., 2021), but may – interestingly – have played a role in driving origination.

While a link between the Shuram excursion and first pulse of extinction at the White Sea–Nama transition is irreconcilable with new radioisotopic constraints, Yang et al. (2021) have provided evidence for a second negative excursion that postdates the Shuram and ended at ~550 Ma. This excursion is potentially correlative with the carbon isotope excursion documented in the basal Nama Group of southern Namibia >548 Ma (Bowring et al., 2007; Wood et al., 2015) and/or negative carbon isotope values documented in the Stirling Quartzite of Death Valley, California (Verdel et al., 2011). The existence of a large negative excursion coincident with the White Sea-Nama transition would invite suggestions of causality, although more data will be required to establish beyond doubt the existence of two large negative carbon isotope excursions in the middle Ediacaran. Furthermore, Eden et al. (2022) found no evidence for a 'catastrophe'-type signature in Nama-aged communities – something that might support a causal relationship with extinction. Nevertheless, the rift-related Central Iapetus magmatic province (CIMP)—with loosely constrained pulses from c. 580–550 associated with the opening of the Iapetus Ocean (Youbi et al., 2020)—provides a prospective mechanism for environmental and/or carbon isotope perturbations in the middle Ediacaran, worthy of further investigation (Figure 3).

With respect to the BACE, although large, negative carbon isotope excursions have been found at horizons approximately coinciding with the E–C boundary on several paleocontinents, there remain significant challenges with correlations and temporal calibration (e.g., Bowyer et al., 2022). Beyond understanding the carbon isotope record, these correlations have significant bearing on how the biostratigraphic records of the E–C boundary interval are interpreted, and thus our understanding of rates and patterns of biotic turnover at this boundary remains limited. Previous calibration of the BACE (and of the E–C boundary) at ~541 Ma hinged on a U–Pb ID-TIMS ash bed date just below onset of the negative carbon isotope excursion in the Ara Group of Oman (Bowring et al., 2007). However, more recent data from Namibia, South Africa, and northern Mexico have revised this calibration (Linnemann et al., 2019; Hodgin et al., 2021; Nelson et al., 2022). Nelson et al. (2022) suggest that, given available age constraints, the BACE and the last appearance of Ediacaran fossils, such as cloudinomorphs and erniettomorphs should be interpreted as <538 Ma. Regardless of its absolute age, as postulated when it was first identified in the 1990s, the BACE continues to hold up as a useful marker of the E–C boundary (e.g., Narbonne et al., 1994; Corsetti and Hagadorn,

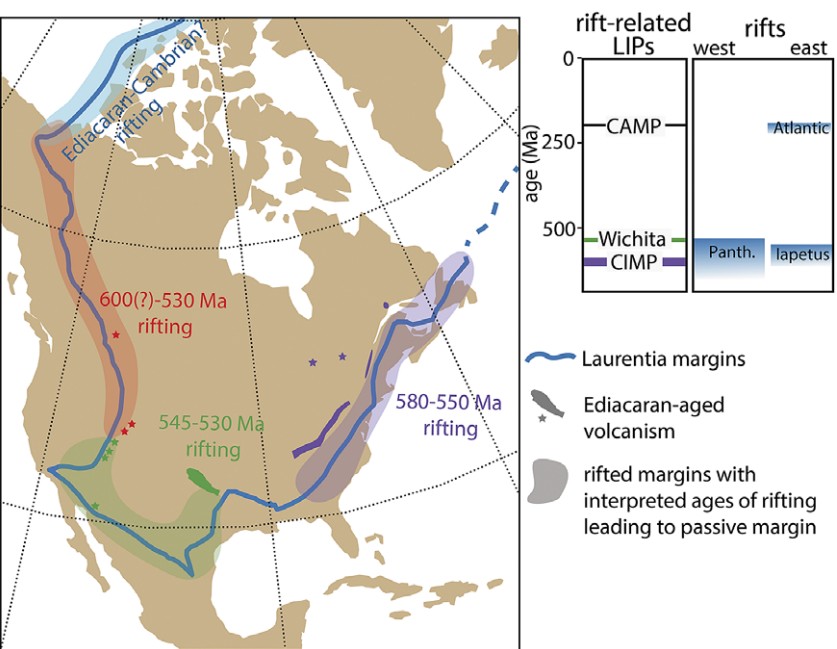

**Figure 3.** Ediacaran–Cambrian rift volcanism in Laurentia and timing of interpreted rifts leading to passive margin development around Laurentia after the breakup of Rodinia. Rifting along the southwestern margin has recently been suggested to coincide with the E–C boundary, potentially resulting in the BACE negative carbon isotope excursion and a second pulse of Ediacaran extinction (see Hodgin et al., 2021; Smith et al., 2022). CAMP, Central Atlantic magmatic province; CIMP, Central Iapetus magmatic province; Panth, Panthalassa Ocean.

2000). Furthermore, more recent data continue to reinforce that the BACE postdated most occurrences of Ediacaran fossils such as erniettomorphs and cloudinomorphs (e.g., Smith et al., 2022) and predated most occurrences of Cambrian trace fossils and small shelly fossils (e.g., Topper et al., 2022), supporting hypotheses of biotic turnover across this excursion.

These issues with correlation feed into discussion of what may have caused the BACE and to what extent it represents a plausible driver for extinction. For example, if emerging geochronological data reinforce the existing difficulties with correlating the BACE across different continents, then this might suggest that it represents a regional or diagenetic signal. We note, however, that the excursion persists across regional sequence boundaries and dolomitization fronts and is thus inconsistent with a purely diagenetic origin (Smith et al., 2022). Assuming that the excursion represents a perturbation to the global marine carbon cycle, Hodgin et al. (2021) suggested a genetic relationship with rift-related volcanism in southern Laurentia. This hypothesis suggests that dissolved inorganic carbon of marine waters attained highly $^{13}$C-depleted compositions due to the combined influx of mantle carbon from volcanic outgassing and combusted organic carbon within intruded sedimentary rift-basins, drawing parallels between end-Ediacaran extinction and the Permian–Triassic and Triassic-Jurassic mass extinction events. While this would be consistent with a 'catastrophe' model for the end-Ediacaran extinction, the proposed mechanistic link remains speculative, as it hinges on temporal correlation between the BACE and pulses of volcanism associated with the c. 539.5–530.0 Ma Wichita igneous province (Hodgin et al., 2021; Wall et al., 2021), as well as stratigraphic correlation to other basaltic and volcaniclastic units in southwestern North America (e.g., Smith et al., 2022; see Figure 3).

This work addressing questions 1 and 2 has naturally fed into question 3 – disentangling correlation vs. causation – in tying the late Ediacaran carbon isotope excursions to potential sources of ecological stress, and thus targeting the 'catastrophe' model for late Ediacaran extinction. However, substantial work has also been done in this vein targeting the 'biotic replacement' model, which suggests that the disappearance of the Ediacara biota was a consequence of ecosystem engineering by the emerging metazoan fauna. Two key questions in this regard have therefore been: (1) how intense was ecosystem engineering in the Ediacaran? And (2) how did the Ediacara biota and metazoan fauna interact?

In terms of the former, much recent effort has focused on the trace fossil record. Not only do bioturbating animals have powerful effects on resource flows and in modifying the physical environment (Rhoads et al., 1978; Jones et al., 1994; Rosenberg et al., 2001; Meysman et al., 2006), but they are also a record of ecosystem engineering that is relatively easy to preserve as fossils (Marenco and Bottjer, 2007). A detailed search through Nama-aged sediments in several localities worldwide has uncovered a remarkable diversity of Ediacaran ichnotaxa (e.g., Parry et al., 2017; Buatois et al., 2018; Turk et al., 2022), while other studies have shown that the impacts of these behaviors potentially reach Cambrian levels millions of years before the E–C boundary (e.g., Cribb et al., 2019). When set alongside the apparent low genus diversity of Nama-aged Ediacara biota (Laflamme et al., 2013; Darroch et al., 2015; 2018a,b), this would seem to lend strong support for the 'biotic replacement' model – at least over the White Sea-Nama transition (i.e., the first extinction pulse). However, there remains uncertainty surrounding the extent to which these ecosystem engineering impacts affected soft-bodied Ediacara biota, and thus to what extent there is a predictable pattern of extinction selectivity (Darroch et al., 2021).

This point dovetails with the latter, centered on ecological interactions between the Ediacara biota and metazoan fauna. The growing recognition that bed-penetrative bioturbation is much more pervasive and extends further back into the Ediacaran than previously thought (e.g., Jensen et al., 2000; Mángano and Buatois, 2020; Nelson et al., 2022; Turk et al., 2022), could suggest that the

removal of microbial matgrounds might have been a plausible source of ecological stress and reduction of taphonomically favorable depositional settings. However, not only is there little evidence that matgrounds disappeared over the course of the E–C transition (Buatois et al., 2014), but in addition, frondose Ediacara biota – potentially the group(s) most reliant on matgrounds – are among the taxa that persist right up until the base of the Cambrian. Other hypotheses surrounding the ecosystem engineering impacts of bioturbation also seem to be inconsistent with patterns of extinction and survival over the White Sea-Nama transition, although are hampered by an incomplete understanding of how many groups of Ediacara biota functioned (Darroch et al., 2021).

Finally, some work has been done in the area of Question 4 ('What role did the end-Ediacaran extinction play in the Cambrian Explosion?'), although principally in terms of framing biotic patterns over the E–C transition as an artifact of evolutionary patterns and stem-vs., crown-group diversity dynamics (i.e., suggesting that Ediacaran extinction and Cambrian explosion are different facets of the same process) – see discussion in the above section: 'Do these intervals of biotic turnover represent extinctions at all?'.

## Key questions in 2023

Arguably, one of the most important questions surrounding Ediacaran extinction events is centered on establishing the magnitude of taxonomic loss over the E–C transition (i.e., to what extent these bioevents represent a 'mass' extinction) – something that will require better biostratigraphic correlations between fossil localities, and more sophisticated methods for extinction rate modeling. However, we argue that there are a number of other guiding questions which will help build more broad understanding for how pulses of extinction may be part of the sustained interval of biotic innovation that helped sculpt the more modern-looking Phanerozoic biosphere. These are summarized below:

### *Were extinction pulses slow or rapid?*

A crucial question that may eventually help distinguish between 'catastrophe' and 'biotic replacement' is whether late Ediacaran turnover pulses were rapid, or more protracted. Recent work on the Phanerozoic 'Big 5' mass extinctions has shown that the majority of these events were geologically rapid, with community collapse in those events unequivocally driven by environmental perturbation (i.e., the Permian–Triassic and Triassic-Jurassic) occurring on the order of $10^5$ years (Burgess et al., 2014; Erwin, 2014). In a 'catastrophe' scenario, therefore, we might expect late Ediacaran turnover pulses to be similarly fast. 'Biotic replacement', in contrast, would intuitively be a slower process, arguably operating over a range of evolutionary and ecological timescales as new organisms/behaviors evolve, become successful, disperse, and finally reach widespread ecological significance. Crucially, this model then predicts lengthy stratigraphic overlap between metazoans and Ediacara biota, with diversity decline beginning either once a key behavior emerges, or a threshold in ecosystem engineering intensity is reached. This point, for instance, stands in stark contrast to claims by Wood et al. (2019) that stratigraphic overlap comprises evidence against the 'biotic replacement' model. In other words, the lengthy co-existence of Ediacara biota and new ecosystem engineering behaviors is something *predicted* by 'biotic replacement', rather than a criterion for rejecting it. One immediate difficulty with answering this question is the paucity of fossiliferous sections

that both span assemblage boundaries and possess sufficient age control. However, recent work on the Dengying Formation in South China has uncovered dickinsoniomorph fossils from near the base of the Shibantan Member (551–543 Ma) below the first occurrence of *Cloudina* (Xiao et al., 2020; Wang et al., 2021). The Shibantan Member thus likely spans the transition between the White Sea and Nama assemblages (Wang et al., 2021), and may offer an opportunity to constrain the timing of a putative first extinction pulse. Recent work from the Nagoryany Formation in Moldova (Francovschi et al., 2021) suggests that these strata preserve a similar interval, and so may offer another opportunity to study this transition in more detail.

### *What were patterns of extinction selectivity and survivorship over the two turnover pulses?*

Extinction events are characterized by 'victims' and 'survivors', the specific identities of which can offer vital clues as to the source(s) of ecological stress and thus help identify the proximal drivers of extinction. To provide a classic example, Knoll et al. (1996) showed over the Permian–Triassic mass extinction that heavily calcified invertebrates with low metabolic intensities suffered considerably higher extinction intensities than other groups; this pattern suggested that hypercapnia (linked to elevated $CO_2$) was a likely culprit – an inference that pre-empted subsequent work establishing LIP volcanism and ocean acidification as overarching extinction drivers. Thus far, relatively little work has gone into analyzing patterns of selectivity over pulses of E–C extinction, beyond several authors noting that sessile, frondose, and semi-infaunal groups of Ediacara biota overwhelmingly survived the first pulse of extinction at the expense of groups that were mobile and/or surficial. Darroch et al. (2021) noted that sources of ecological stress associated with the specific impacts of bioturbation were hard to ally with the observed extinction selectivity patterns, although there remain big knowledge gaps surrounding what the ecosystem engineering impacts of early metazoans actually were (see question 4 below). In one of the few studies to directly address this question of selectivity, Evans et al. (2022) have noted that the overwhelming survivors of the first extinction pulse – the rangeomorphs and erniettomorphs – are characterized by high surface-area to volume ('SA:V') ratios (Laflamme et al., 2009). With the assumption that high SA:V ratios represent an adaptation (or advantage) to surviving in low-oxygen conditions, then the observed pattern of survivorship would be consistent with extinction driven by fluctuations in global redox conditions (see also Evans et al., 2018). We note that there are ambiguities surrounding to what extent the erniettomorphs had high surface areas exposed to the water column in life – most seem to have lived at least partially buried in the sediment (Ivantsov et al., 2016; Gibson et al., 2019; Darroch et al., 2022) – and whether their body plans were evolved for gas exchange as opposed to feeding (see Laflamme et al., 2013). Despite these caveats, Evans et al. (2022) demonstrate that focusing on selectivity is a powerful means for hypothesis testing, and a more complete knowledge of the physiological, paleoenvironmental, and paleoecological characteristics of 'victims' and 'survivors' over this boundary could offer vital hints as to what may have driven the first extinction pulse.

The second extinction pulse is similarly enigmatic. Although the vast majority of Ediacara biota disappear at, or shortly beneath, the base of the Cambrian worldwide, there are tantalizing hints that some taxa may persist for short intervals into the lower Cambrian (although we note the difficulties with defining the base of the Cambrian in some of these key sections – e.g., Nelson et al.,

2022). However, there have been no convincing Ediacara biota reported from younger sediments, suggesting that survivors may be 'dead clades walking' – a characteristic of several Phanerozoic extinctions whereby some species persist in low numbers through an extinction only to die out early in the recovery interval (Jablonski, 2002; Hull et al., 2015). Among putative survivors, three examples stand out as being particularly convincing and/or in need of closer analysis: Jensen et al. (1998) figure apparent erniettomorph Ediacara biota from the Uratanna Formation in South Australia, stratigraphically above the FAD of *T. pedum*. In the southwestern United States, Hagadorn and Waggoner (2000) report forms similar to the erniettomorph taxon *Swartpuntia* from two separate localities above the FAD of *T. pedum*, and from strata that also preserve trilobites and archaeocyaths. Most recently, Nelson et al. (2022) report an erniettomorph from the Nomtsas Formation in the Neint Nababeep Plateau in South Africa. Although they interpreted this occurrence as Ediacaran, given that the Nomtsas Formation is traditionally interpreted as Cambrian (e.g., Linnemann et al., 2019), this may represent yet another survivor. Although these data are sparse (and have yet to be fully investigated beyond an initial description), a preliminary analysis of selectivity across this second extinction pulse may suggest that frondose erniettomorphs were survivors, while rangeomorph Ediacara biota were permanent casualties (although see Hoyal Cuthill, 2022 for an alternative view of rangeomorph extinction). If it is shown that erniettomorphs had significantly different paleobiologies or -ecologies than these other groups, then this may help establish extinction drivers over the E–C boundary itself.

### Are Ediacaran–Cambrian carbon isotope excursions recording global environmental perturbations?

Carbon isotope excursions in the late Ediacaran and earliest Cambrian are, largely, interpreted as recording perturbations to the global dissolved inorganic carbon (DIC) reservoir. As such, $\delta^{13}C$ chemostratigraphy from carbonate successions has been used to correlate between sites regionally and globally, to construct age models, and to calibrate biostratigraphic changes across the E–C transition, particularly in the absence of radioisotopically constrained sections (e.g., Maloof et al., 2010; Bowyer et al., 2022). Although this approach has been applied widely to E–C studies, a number of geochemical studies of modern and ancient carbonate platforms have demonstrated the pitfalls of indiscriminate use of carbon isotope chemostratigraphy. Studies of modern carbonate platforms have shown that mineralogy and early marine diagenesis in shallow marine environments can result in variable $\delta^{13}C$ records that are recording sediment- and fluid-buffered diagenetic precipitates that can be decoupled from global DIC (Swart, 2008; Oehlert and Swart, 2014; Higgins et al., 2018). Building upon these modern studies, chemostratigraphic data from Neoproterozoic carbonate platforms have demonstrated lateral $\delta^{13}C$ variability across shelf to slope transects and, using the interpretive framework established in studies of modern carbonate platforms, have interpreted some of this variability as the result of early diagenesis (Ahm et al., 2019; Hoffman and Lamothe, 2019). Other studies of partially dolomitized Neoproterozoic platforms have demonstrated that dolomitizing fluids have the potential to alter $\delta^{13}C$ values by up to 10‰ (Bold et al., 2020; Nelson et al., 2021). Finally, some detailed stratigraphic investigations have demonstrated a facies dependence on the character and/or preservation of carbon isotope excursions (e.g., Lu et al., 2013; Busch et al., 2022). Collectively, these studies highlight the need for careful assessment of the diagenetic histories of individual E–C carbonate platforms before records from individual sites can be interpreted within a global framework. With the recognition of more E–C carbon isotope excursions, some of which are only convincingly documented in a single region (see Figure 1), this regional scale assessment of diagenetic history is particularly important.

A parallel challenge for the community will be linking individual negative carbon isotope excursions that *are* established as global, to viable environmental perturbations that could result in an influx of light carbon. This challenge is not a new one. The extreme fluctuations in $\delta^{13}C$ that characterize much of the Neoproterozoic have long been difficult to interpret in a mass-balance framework because of the dramatic changes in oxidants that are implied. Despite these long-standing challenges, rifting and rift-related volcanism of broadly E–C age occurred around the margins of Laurentia (Figure 3) and, as with some of the volcanic episodes associated with Phanerozoic mass extinctions, have recently been proposed as a possible "trigger" for a cascade of E–C environmental, ecological, and biotic effects (Hodgin et al., 2021; Smith et al., 2022). Similar to the historical trajectory of the study of Phanerozoic mass extinctions (e.g., Newell, 1967), the first step in testing the idea that E–C rift-related volcanism caused a perturbation(s) to geochemical cycles and a biotic crisis is demonstrating temporal coincidence among them. After temporal coincidence is established, the focus can turn to studying the geochemical response and the expected biotic selection across the E–C transition.

### What were the ecosystem engineering impacts of early animals?

The question as to how Ediacara biota and metazoans were interacting (discussed above) has yet to be fully answered, however, there are arguably more fundamental questions surrounding what the ecosystem engineering impacts of early animals actually were, and whether they were capable of driving large scale environmental change. While Cribb et al. (2019) focused on trace fossils and quantified their effects as indices of 'ecosystem engineering impact' (the 'EEIs' of Herringshaw et al., 2017), these may overestimate downstream effects, and so their use has been criticized (Minter et al., 2017). Other workers have highlighted the importance of biomixing vs. bioirrigation. For clarification, biomixing has relatively little ecosystem engineering impact (especially at shallow depths), whereas bioirrigation is a more powerful driver of environmental change – leading to deepening redox gradients, altered distribution of redox-sensitive elements, and increased availability of organic matter (e.g., Kristensen et al., 2012; Tarhan et al., 2015, 2021; Darroch et al., 2021). In this regard, the presence of treptichnid burrows in the late Ediacaran (Cribb et al., 2019; Jensen et al., 2000; Darroch et al., 2021) is crucial; treptichnids record the first substantial bioirrigative behaviors to appear in the trace fossil record (likely between 542.65–539.63 Ma; see age model in Nelson et al., 2022) and rapidly increase in size, complexity, and intensity through the latest Ediacaran and into the Cambrian. However, the actual downstream effects of this style of burrowing have not been measured, and so the significance of Ediacaran treptichnid-like behaviors is unknown. What is needed is a combination of in vivo ichnological experiments (encompassing a wide variety of different tracemakers) with integrated geochemical models; with these two approaches we might reasonably hope to understand what the significance of these behavioral innovations may have been (see, for example, Cribb et al., 2023).

Lastly, in addition to the record of bioturbation, there is a wealth of other ecosystem engineering impacts that appear at approximately the same time and should be more broadly considered (e.g., Erwin and Tweedt, 2012). Recent studies using fluid dynamics modeling have noted a paleoecological shift in the prevalence and character of inferred suspension feeders from the White Sea into the Nama intervals (Gibson et al., 2019;Cracknell et al., 2021; Darroch et al., 2022), which could have altered resource flows, and the distribution of habitable ecospace. Alternatively, it is also becoming apparent that passive predation is a mode of life that perhaps emerged as early as the Avalon assemblage (Liu et al., 2014; Dunn et al., 2022), but then may have expanded dramatically in the Nama (Bengston and Zhao, 1992; Darroch et al., 2016; Schiffbauer et al., 2016; Leme et al., 2022; Turk et al., 2022). Although not strictly an ecosystem engineering impact, the rise of predation was likely a source of ecological antagonistic stress that could have gradually marginalized the Ediacara biota, particularly if any of these groups possessed a mobile larval or dispersal stage early in development (Darroch et al., 2016). A prediction of this might be that Ediacaran communities from the Avalon through Nama assemblages show a noticeable shift from more neutral to niche-dominated processes in ecosystem dynamics – a facet of community paleoecology that can be readily preserved in the spatial distributions of fossils on bedding planes (see Mitchell et al., 2019, 2022).

### *Bonus question: What were the ecosystem engineering impacts of Ediacara biota?*

Finally, much discussion surrounding the 'biotic replacement' model has focused on the impact of metazoans and the emerging Cambrian-style fauna as ecosystem engineers. However, this conceptual approach ignores to what extent the evolving Ediacara biota may have been engaged in forms of ecosystem engineering themselves, and thus may be reinforcing a false narrative surrounding the character and drivers of turnover. Although none of the Ediacara biota are thought to have disrupted the sediment–water interface to the extent that Cambrian-style metazoans did, many might have engaged in other forms of ecosystem engineering. For example, as mentioned above, several groups of Ediacara biota were likely suspension feeders (Rahman et al., 2015; Gibson et al., 2019) and thus may have helped fuel the Cambrian explosion through forging energetic links between pelagic and benthic realms (Cracknell et al., 2021; Darroch et al., 2022), as well as help oxygenate the water column (Erwin and Tweedt, 2012). In addition, other groups of sessile – predominantly frondose – Ediacara biota apparently formed dense 'meadows' in both shallow and deep-water settings (see, e.g., Clapham et al., 2003; Droser and Gehling, 2008) which would have baffled currents, altered resource flows, and created a diversity of benthic niches and hydrodynamic refugia that could have been exploited by other taxa (analogous to modern seagrass meadows – see, e.g., Gartner et al., 2013). Lastly, Budd and Jensen (2017) have suggested that, following death and decay, patches of sessile Ediacara biota may have served as rich sources of organic matter – similar to whale falls – that in turn would have provided a selective pressure towards motility and the development of deposit-feeding strategies. Although some of these models are more plausible than others, they collectively emphasize that little is currently known about the ecosystem engineering impacts of the Ediacara biota. More broadly, these ideas illustrate that a key facet of understanding pulses of late Ediacaran extinction – both in terms of testing

between hypothesized drivers and analyzing extinction selectivity – is understanding what different groups of Ediacara biota were actually doing within their ecosystems.

## Summary

Pulses of extinction in the late Ediacaran remain among the most enigmatic events in the history of life – occurring at a crucial interval during the Neoproterozoic rise of animals – and thus potentially influencing trends in early animal evolution as well as the character of the emerging, more modern-functioning and animal-dominated marine biosphere. The extent to which these events represent 'mass extinctions' – that is, the rapid disappearance of >70% of marine genera, rather than less severe and more protracted turnover pulses, is a question that doubtless requires a more focused analysis. Inferring the drivers of these putative extinction pulses is also a question that is fraught with difficulties stemming from biostratigraphic correlation and the interpretation of enigmatic geochemical signals. Despite this, the last 5 years of research has produced a wealth of new geological, paleontological, geochemical, and geochronological data that are slowly bringing this interval into focus. These new data have answered several of the questions posed by Darroch et al. (2018a), but have left others unanswered. Moreover, this work has led to new questions which promise to not only help unravel this critical interval in Earth's history, but also contribute to our knowledge of extinction events more generally. Addressing the questions listed above will: (1) help link sources of environmental change in the late Ediacaran with drivers of ecological stress; and (2) explain patterns of extinction and survivorship across extinction pulses. Taken together, this new information will allow for a coherent picture of late Ediacaran extinction and help determine the role it may have played in the Cambrian explosion.

Finally, the last 5 years of research allow us to propose a revised hypothesis for drivers of E–C biotic turnover, that can be tested and refined with further discoveries. Modeled after Knoll and Carroll (1999), the various tenets of this hypothesis build off a combination of growing consensus and current ideas, specifically: (1) the majority of 'Ediacara biota' are stem-group members of extant animal phyla, with crown group members of these same clades emerging as early as the Avalon assemblage (e.g., Dunn et al., 2018; Dunn et al., 2022); (2) in the absence of temporal correlation between isotope excursions and diversity loss (Rooney et al., 2020), a first pulse of extinction is recognized at the White Sea-Nama transition driven by competition, biotic interactions, and widespread geobiological change stemming from the diversification of crown-group metazoan clades (i.e., 'biotic replacement'; see Darroch et al., 2015; Schiffbauer et al., 2016); and (3) a second pulse of extinction at the E–C boundary, potentially driven by widespread environmental perturbation (i.e., 'catastrophe') following extensive rift volcanism around the southern and southwestern margins of Laurentia (Hodgin et al., 2021; Smith et al., 2022) (Figure 4). This model thus hypothesizes a two-pulsed extinction of the Ediacara biota – with a first pulse driven by 'biotic replacement' and a second pulse driven by 'catastrophe' (i.e., environmental perturbation), combining recent evidence from paleontology, geochronology, and geochemistry. This model is obviously highly speculative and sensitive to the questions outlined above, but does frame testable hypotheses that can, and will, be targeted in the next decade of work. Testing this model will allow us to fold the biotic turnover events occurring over the E–C transition into broader discussions surrounding the tempo,

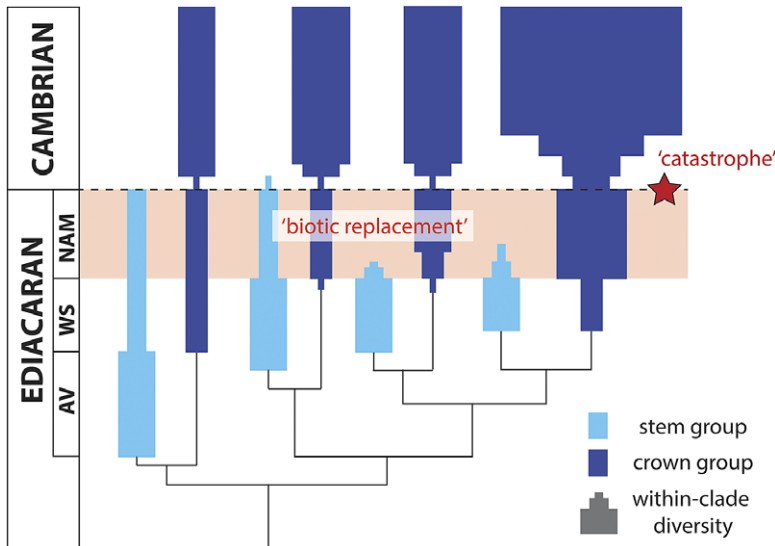

**Figure 4.** A hypothetical model for drivers of the E–C transition, modeled after Knoll and Carroll (1999); their figure 5). 'AV' = Avalon; 'WS' = White Sea; 'NAM' = Nama. This model interprets the majority of 'Ediacara biota' as stem-group metazoans, with a first pulse of extinction at the White Sea-Nama transition driven by the diversification – with associated downstream geobiological impacts – of crown-group metazoan clades (i.e., 'biotic replacement'). A second pulse of extinction follows at the E–C boundary, driven by widespread environmental perturbation (i.e., 'catastrophe') following extensive rift volcanism. We note that this figure is strictly hypothetical, and the stem – and crown-groups depicted here do not intentionally correspond to specific biological groups shown in Figure 1.

mode, and drivers of mass extinction events, and invite comparisons with the 'Big 5' mass extinctions of the Phanerozoic.

**Open peer review.** To view the open peer review materials for this article, please visit http://doi.org/10.1017/ext.2023.12.

**Supplementary material.** The supplementary material for this article can be found at https://doi.org/10.1017/ext.2023.12.

**Acknowledgements.** S.A.F.D. and M.C. are grateful to the Ecological and Evolutionary Effects of Extinction and Ecosystem Engineers ('E6') group for helpful discussions and feedback. We thank our reviewers and handling editor for thoughtful comments and discussion surrounding E–C extinction rates, and which considerably improved an earlier version of this manuscript.

**Author contribution.** Conceptualization: all authors.; Formal analysis (principally Figure 1 and calculation of extinction intensities): M.C.; Funding acquisition: S.A.F.D., E.F.S., M.L. and J.D.S.; Writing—original draft: S.A.F.D.; Writing—review & editing: all authors.

**Financial support.** S.A.F.D. was supported by a National Science Foundation RCN grant (NSF-DEB 2051255), and joint funding from the National Science Foundation (NSF EAR-2007928) and Natural Environment Research Council (NE/V010859/2). S.A.F.D. also acknowledges generous support from the Alexander von Humboldt Foundation, which is sponsored by the Federal Ministry for Education and Research in Germany. M.L. was supported by an NSERC Discovery Grant (RGPIN 435402). E.F.S. acknowledges support from the National Science Foundation (NSF EAR-2144836 and NSF EAR-2021064), the Sloan Research Fellowship (#FG-2021-2116,049), and the Johns Hopkins Catalyst Award. This is E6 (Ecological and Evolutionary Effects of Extinction and Ecosystem Engineers RCN) publication #2. J.D.S. acknowledges support from the National Science Foundation (NSF EAR CAREER-1652351).

**Competing interest.** The authors declare none.

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
