## [Reviewer Report]

Unsurprisingly, this manuscript is about extinction, which is the ultimate fate of all taxa, just as death comes eventually to all individuals. So the question is, were the extinctions of Ediacaran taxa synchronous enough to move a normal process into the abnormal range of mass extinction and thus warrant this protracted discussion?

Here, the reliability of the observed fossil record becomes paramount. Are the Avalon, White Sea and Nama assemblages really separated by mass extinctions or are they windows into an evolving biota where change seems abrupt because of intermittent viewing? It is circular reasoning to think that the White Sea and undated South Australian assemblages are the same ago because they so similar, and then to use the fact that they are so similar to argue that the known duration of the White Sea assemblages, based on U-Pb ages, implies that the South Australian examples went extinct at the same time.

Furthermore, the Nama assemblage consists of a few distinctive taxa (Pteridinium, Rangea, Archaeichnium, Ernietta, Swartpuntia) that—at best—are known from rare specimens elsewhere. In South Australia, where the succession is clear and the sampling outstanding, the Nama taxa Pteridinium and Rangea are only found beneath the beds that contain the ‘White Sea’ assemblage despite almost a century of targeted collecting.

Line 49: 1980s not 1980’s.

121-123: Whether the Ediacara biota were metazoans or not has no bearing on their propensity for extinction. In any case, the Ediacara biota has already been segregated from the Metazoa (102) because they lack the body plans of the animal phyla.

133-136: There has been a lot of water under the bridge since Amthor et al (2003) and the results they presented are probably incorrect about the age of BACE and the extinction of cloudinids. In any case, losing a couple of calcareous taxa (Cloudina and Namacalathus) is hardly grounds for a mass extinction. The authors seem to confuse fossil abundance with taxonomic diversity.

137-153: Everyone agrees that there are more Ediacara biota taxa in the White Sea assemblages than in the Nama assemblage, but is this because of extinction or loss of information? Apart from the famous Pteridinium locality at Aar farm, Ediacaran fossils are rare in Namibia and only one or two modes of preservation are found. Most importantly, there are very few of the fine bed surfaces that are so common at numerous White Sea and South Australian localities. With this background, it is not helpful to use the ‘Kotlinian crisis’, which - as the authors acknowledge - was introduced for a fundamentally different reason to support their purported White Sea-Nama mass extinction event.

174-176: ‘genus richness among the erniettomorphs etc.’ This doubtful argument is based on a small number of rare fossils (both taxa and specimens). Ernietta is known from only two narrow stratigraphic intervals, one in Namibia and one in Nevada. Swartpuntia, another genus that increases the biodiversity of the Nama assemblage, perhaps enough to tip the balance, is known from only one place on the planet if doubtful identifications in Nevada, Australia and North Carolina are disregarded, as they should be for discussions of this nature.

190-198: The evidence presented for extinction at the Ediacaran-Cambrian boundary suffers from another difficulty: taxonomic splitting. It is indisputable that ‘worm tubes’ and cloudinids are common fossils in the latest Ediacran, but anyone who has worked on small shelly fossils from the early Cambrian would know that tube fossils did not go extinct. They just became less important by dilution with other more interesting, shelly fossils. Because tubular fossils are so rare in the Proterozoic they have been studied and named far more thoroughly, so counting the loss of generic names across the boundary is not necessarily good evidence for mass extinction.

266: ‘followed by the origin of new clades’ is not correct. Most post-extinction expansive clades (angiosperms, birds, mammals, ammonites) originated prior to the extinction event. They may have radiated spectacularly afterwards but they did not originate after the event.

313-314: Add a reference to Topper et al. (2022) when discussing the timing of the BACE.

Fig. 1. This figure is misleading in that it implies that the sudden originations and expansions at the beginnings of the periods represented by Avalon, White Sea and Nama are based on evidence from the fossil record. There is no evidence for synchronous origins (or extinctions) of these putative clades. Their stratigraphic ranges should be shown as bars with fuzzy beginnings and endings. Leaving out the horizontal lines, which imply a precision that does not exist, might help. For example, even if Kimberella is a bilaterian - which many dispute - there is no evidence that its origin coincided with the first appearance of other members of the White Sea assemblage. Trace fossils may be a more useful guide to the time of appearance of bilaterians. Similarly, the presence of Porifera and Cnidaria in the Avalon assemblage is by no means certain, so it is even more problematical to postulate that both phyla appeared synchronously at the beginning of Avalon time. If these uncertainties are properly displayed, the case for mass extinction can more easily be evaluated by the readers. Also, Porifera (spicular sponges and archaeocyaths) underwent a spectacular radiation during the early Cambrian, not shown on this figure.

Fig. 4. Avoid following Knoll and Carroll (1999) for the model shown in Figure 4. There has been a huge increase in knowledge from genomics since then and most biologists would now place sponges and their grade of organization at the base of the metazoan tree. If so, almost all members of the Ediacara biota become crown not stem metazoans, as discussed earlier in the manuscript. This problem could be finessed in Fig. 4 by speaking about stem and crown eumetazoans, but the branch order in the stem group then needs to be addressed and made congruent with the clades shown in Fig. 1. At present, these two figures present very different views of the Ediacaran history of the Metazoa. In Fig. 1, the metazoan crown group originates prior to Avalon time, the Bilateria and presumably the Lophotrochozoa prior to White Sea time, and six Ediacara Biota/stem Eumetazoa groups originate synchronously at the Avalon-White Sea boundary, as shown by their point sources.

Also, where are the data that would support a bottleneck for the crown taxa at the E-C boundary? The worm tubes? Cloudina (arguably a metazoan) and Namacalathus (probably not)? According to Fig. 1 all metazoan groups except worm tubes, which in any case are only form taxa, were diversifying at this time.

In summary, the case for mass extinctions at the end of White Sea time and at the end of the Ediacaran is not strong. Although widespread negative carbon isotope excursions (BANE and BACE) may have occurred at the same times, there is no evidence apart from temporal coincidence, that these excursions were involved in the proposed extinction events. There is a need to address the discrepancies between Figs. 1 and 4. Fig. 3 could be omitted as it is barely discussed and only applies to Laurentia.

---

## [Reviewer Report]

This manuscript reviews current thinking and recent research regarding proposed mass extinction events in the latest Ediacaran Period, before posing a series of new questions that could provide a framework for future research in this area. It is a well written and coherent summary of this topic, and addresses several pertinent issues. There are undoubtedly other questions that could be considered, but the article works as a succinct and approachable overview of current thinking. I have suggested a few places where additional references could be considered, and one potential edit to Figure 4, but I have no major concerns, and I congratulate the authors on producing an enjoyable and informative paper.

The authors may want to consider discussion/inclusion of the recent Evans et al (2022) paper in PNAS, published after submission of this manuscript:

Evans, Scott D., et al. “Environmental drivers of the first major animal extinction across the Ediacaran White Sea-Nama transition.” Proceedings of the National Academy of Sciences 119 (2022): e2207475119.

Minor comments:

l. 96: There is scope to refer to biomarker evidence for metazoans here too (Bobrovskiy et al 2018 Science paper on Dickinsonia, or their very recent 2022 Current Biology paper on Kimberella and Calyptrina diet). It might also be relevant to mention the trace fossil record early here, to demonstrate the co-existence of latest Ediacaran biotas with bilaterian trace makers.

l. 153: The “Kotlinian” is time-equivalent to this crisis, as discussed in Grazhdankin (2014) (Patterns of evolution of the Ediacaran soft-bodied biota. Journal of Paleontology, 88(2), 269-283).

l. 166-169: This statement seems quite ‘certain’, given the uncertainty surrounding timings and ages of many global sites (e.g. South Australia), and recent discoveries (such as dickinsoniid specimens in the ?Nama-aged Shibantan Mbr: Wang et al., (2021), as you mention later).

l. 198: If mentioning the possibility of survivorship of Ediacaran taxa across the boundary, it is probably worth noting the existence of candidate soft-bodied ‘survivors’ too for completeness (recently summarised in Hoyal Cuthill, 2022: Ediacaran survivors in the Cambrian: suspicions, denials and a smoking gun. Geological Magazine, 1-10; or this very recent paper: Hu et al (2022). A new Cambrian frondose organism:“ Ediacaran survivor” or convergent evolution?. Journal of the Geological Society, jgs2022-088).

l. 248: Perhaps cite some additional examples for these points (as you have done for the Shuram above).

l. 261-263; Arguably there was/is minimal independent geological evidence for a catastrophe either (in the form of LIPs, impacts, OAEs or any of the other usual suspects correlative particularly with the older event).

l. 288: Matthews et al 2021, not 2020

l. 366: Mangano and Buatois 2020 also seems relevant as a review of this extended trace fossil record here: Mángano, M. G., & Buatois, L. A. (2020). The rise and early evolution of animals: where do we stand from a trace-fossil perspective?. Interface Focus, 10(4), 20190103.

l. 407: This is a valuable point, but it would be more informative to a reader if the rationale provided for those counter-claims by Wood et al (2019) were summarised here, and then elaborated on to present how, if at all, those alternative interpretations feed into the question of whether the extinction pulses were rapid or slow. Unfortunately I fear that even if the Shibantan presents an opportunity to track evolution across this interval, determining whether the transition is fast or slow will require high-precision dating within that succession, which may be difficult/impossible to achieve without methodological advances.

l. 435: On this point, better constraint of palaeolatitudinal characteristics would also be welcome, and has recently been attempted by both Evans et al 2022 (above), and Boddy et al 2022: Palaeolatitudinal distribution of the Ediacaran macrobiota. Journal of the Geological Society, 179(1).)

l. 445: Jensen et al 1998 discuss their specimens as ‘Swartpuntia-like’ rather than arboreomorphs. I’m not aware of any reassessment of the specimens since? If I’m correct here, Figure 1 may require editing.

l. 479: Studies such as Lu et al (2013; The DOUNCE event at the top of the Ediacaran Doushantuo Formation, South China: Broad stratigraphic occurrence and non-diagenetic origin. Precambrian Research, 225, 86-109.) demonstrate a facies dependence for the nature of CIEs in the Neoproterozoic too, which is of relevance here.

l. 514: ‘treptichnids record…’? Rather than treptichnids are.

l. 535 (see also l. 548): Discussion of some of the work Emily Mitchell is doing on palaeoecology might be relevant here (e.g. Mitchell et al 2019, The importance of neutral over niche processes in structuring Ediacaran early animal communities. Ecology letters, 22(12), 2028-2038, or her recent trace fossil paper Mitchell, E. G., Evans, S. D., Chen, Z., & Xiao, S. (2022). A new approach for investigating spatial relationships of ichnofossils: a case study of Ediacaran–Cambrian animal traces. Paleobiology, 1-19.)

l. 966: How were these extinction intensities calculated?

Figure 1: I’m interested to know more about the basis of the data behind the shapes of the Cnidaria and Porifera diversity curves here (what exactly are the sponges in the White Sea assemblage, and do these plots account for basal/early Cambrian sponge taxa discovered from the Yanjiahe Fm, Soltanieh Fm, or the Hetang biota?) Which taxa are being interpreted as cnidarians in the Nama and earliest Cambrian settings? (and if you’re referring to tubular taxa as candidate cnidarians, are they being counted twice here?)

Figure 2 caption: please include information on where these specimens are from (locality and stratigraphy), along with any associated museum accession numbers (the Australian specimens in particular should include any SAM P numbers).

Figure 4: This figure works as a simple conceptual model, but an alternative version could envisage the Ediacaran biota including stem group members of multiple extant metazoan (and algal/protistan) clades that had already diverged prior to either extinction pulse. So, rather than one lineage crossing the boundary, I would envisage multiple lineages crossing the boundary, each with their own stem groups (e.g. what you’ve drawn being repeated for cnidarians, sponges, and bilaterians at least as individual clades within the Metazoa, in addition to algae and others. Potential stem group metazoans, stem group eumetazoans, and stem groups to individual metazoan phyla could also be truncated by the extinction events, or cross them). This adds complexity, but I think is more phylogenetically correct given current understanding.

---

## [Reviewer Report]

I enjoyed reading the manuscript ‘Causes and consequences of end-Ediacaran extinction – an update’, submitted to Cambridge Prisms: Extinction by Darroch et al. I think that the paper will be a valuable resource and framework for future work on the Ediacaran–Cambrian transition. Please find my review below:

This review-style paper of the Ediacaran-Cambrian transition serves as a 5-year update of Darroch et al. (2015). There are many new and interesting discussion points, in particular about patterns of extinction selectivity. The synthesis is valuable in re-evaluating the possibility of extinction occurring in the late Ediacaran and at the Ediacaran-Cambrian boundary. The review is prescient given that it has recently been shown that there is considerable uncertainty in the timing of the Ediacaran-Cambrian transition. The paradigm of two extinctions with the first driven by biotic replacement and the latter driven by environmental perturbation is a useful framework. The review paper serves as a fresh forward-looking synthesis that can help enliven debate in upcoming years, and serve as a reference point for considering the role of extinction events leading up to the ‘Cambrian Explosion’. Minor revisions are suggested for the text and figures. The minor revision suggestions can be found below amongst the line edits:

Line 55 — …several of the authors listed…

Lines 66-77 — consider adding references when introducing terminology such as “Rise of animals” or the “Big 5”

Line 157 — add a period

Line 189 — Following this summary, perhaps make a summary statement about the likelihood of late Ediacaran extinctions events.

Lines 231-41 — This may be one of the key questions posed in the 2018 paper, yet it is not clearly motivated here. It is not immediately obvious why the Shuram excursion is relevant to End-Ediacaran extinction? Perhaps an additional sentence of introduction/motivation would be helpful to map out the linkage (e.g., explicitly discussing its potential relationship to an origin of Ediacara biota or to an Ediacaran extinction event or that it could be an analogue to the BACE due to the magnitude of the excursion?).

Lines 284-88 — There are other excursions between the Shuram and the BACE… okay, I see this is mentioned in the next paragraph.

Lines 303-04 — It may be worth mentioning Verdel et al (2011) and the excursion in the Stirling Quartzite.

Lines 340 — Given the age constraints from Nelson et al (2022) and the suggestion that the BACE and the basal Cambrian may <538 Ma, it could be worth adding a sentence about the younger ages of the more voluminous portions of the Wichita Large Igneous Province.

Line 469 — Change ‘or’ to ‘of’

Line 485 — negative carbon isotope excursions

Line 498 — across the E–C transition.

Line 519 — no hyphen needed, in vivo

Line 576 — Earth history (no capitalization necessary of history)

Lines 585-94 — Consider making this one sentence using semi-colons (…specifically: (1) ... ; (2) … ; (3).)

Line 589 — Can the White Sea-Nama transition be considered a boundary?

Line 589 — ‘pulse of extinction is recognized at the White Sea-Nama boundary

Line 594 — There is a lot of information within the three-point summary. Prior to the final sentence inserted on line 594, it might be good to add a more condensed summary in which you refer to Fig. 4. Maybe something like: “To summarize, there are two apparent stages of extinction with the first driven by biotic replacement and the latter driven by environmental perturbation (Fig. 4).”

It might be worth reinforcing key concepts brought up in the introduction, such as extinction leading up to and at the E-C boundary being potentially classified along with the ‘Big 5’.

Line 966 — % age, why not write out percentage?

Line 971 — bioirrigation

Line 983 — Why is the Columbia River Flood Basalt included in the figure? It isn’t obvious that it’s relevant as point of comparison, being unrelated to rifting or extinction and being temporally far-removed from the late Ediacaran to E-C boundary.

Figures — Only figures 1 and 3 are referred to in the main text.

Figure 3 — i) On the NW margin of Laurentia, what is the timing of rifting referred to as “Ediacaran-Cambrian rifting” when numeric age ranges are given for the other margins?

ii) There is some agreement in color between panel a and panel b, such as purple = CIMP, green = Wichita. Should there be some sort of legend or explanation for the colors on the rift-related colors on the western and northwestern margins?

iii) Why is the CRFB shown? It does not appear on the map. It doesn’t appear to serve a purpose and it is not referred to in the main text.

Figure 4 — Explain what AV, WS, and NAM stand for.

---

## [Editor Report]

Dear Authors,

Thank you for submitting your manuscript to our journal. Based on the evaluations provided by three expert reviewers and my own reading of the manuscript, your submission requires substantial revisions before it can be accepted for publication. The most critical review (’Major Revisions‘) raised multiple important points that need to be addressed. In addition, the other two reviewers -- despite recommending ’Minor Revisions‘ and ’Accept', respectively -- provided numerous useful suggestions that should be addressed carefully as well.

When returning the revised manuscript, please make sure to provide a detailed rebuttal letter explaining all your responses to the reviewers' comments.

Thank you for submitting your work to our journal,

Michal Kowalewski

---

## [Reviewer Report]

Reviewer 1 is still not convinced, even with an adequate knowledge of the recent literature. However, as this is a solicited review they are willing to give this manuscript a passing grade, but strongly recommends that the matters detailed below be addressed.

As the figures are bound to be the most influential parts of this paper, let’s deal with them first.

Fig. 1. It is difficult to believe that the Avalon-White Sea transition, which for some reason is regarded as a mass origination rather than a mass extinction, represents the evolutionary appearance of nine major clades, but that’s what this diagram implies. If the vertical bars were known stratigraphic ranges, the figure would be very different. To take an extreme example, the Pentaradialomorpha is known from only one undated site in South Australia. More abundant groups, such as the Erniettomorpha, may have the stratigraphic range shown in Fig. 1, but the range extension into the Cambrian (lines 535-537) is based on two inadequately documented reports, one of which may be Ediacaran not Cambrian and the other not an erniettomorph. Furthermore, it is hard to believe that the origin of the Erniettomorpha would coincide exactly in time with the origin of the Bilateria, whether that is based on body fossils or trace fossils, but that’s what a literal reading of this figure requires. Furthermore, the oldest known erniettomorph (Pteridinium) occurs beneath a White Sea ash dated at 552.85±0.77 Ma so extending the range of the group to 560 Ma is a stretch, but it should not be presented as the exact time of origin of the whole clade. If these kinds of uncertainties were incorporated into this figure, it would be a better representation of the state of the art. Perhaps this figure should be totally recast using, for example, 95% confidence intervals on stratigraphic ranges?

Fig. 3 and the text that goes with it (lines 374-388) are merely a distraction. The negative carbon isotope excursion at ca. 500 Ma may exist in China but its meager expressions in Namibia and California-Nevada need to be confirmed before they can correlated with the poorly dated CIMP event, let alone form the basis for an extinction hypothesis. Suggest that these items be removed.

They are also confusing in that the ‘catastrophe’ that serves as a Deus ex machina for the BACE excursion/end Ediacaran extinction in Fig. 4 is attributed (unrealistically) to the Witchita igneous province and some approximately coeval volcanics in the American southwest (lines 421-425), not to the CIMP volcanism.

Fig. 4 has been redrawn to avoid conflict with the range chart of Fig. 1, but the interested reader will try to make the comparisons anyway. It seems that the alternate light and dark blue clades correspond, from left to right, to the Rangeomorpha, stem to the Porifera and/or Cnidaria; Erniettomorpha, stem to an unknown group: Dickinsoniomorpha, stem to another unknown group; and Bilateriomorpha, stem to the Bilateria (including trace and tube fossils). This is a novel reading of the record but it may be correct. However, there is no evidence from the data presented in Fig. 1 for a substantial bottleneck in the bilaterian clade, unless it is the wine glass-shaped depiction of the history of the ”Tubefauna”. The equivalent notches in the other three crown clades have even less support, since the crown group members of the Erniettomorpha and Dickinsoniamorpha have not been identified. The ‘catastrophe’ is largely artistic license.

The other problem with this figure is that it promotes the view that stem groups are doomed to fail by their very nature. This is how many paleontologists viewed the pre-extinction history of the dinosaurs prior to Alvarez et al. They were already in decline because they knew that the end of the Mesozoic was coming. More realistically, stem and crown lineages are identical at the time they split and it is only which one that survives that defines both. It is better to represent stem lineages as many equal branches of a generally diversifying tree with random odds of making it to the next level. Thus, Ediacaran lineages of crown groups are no different in principle from their sister stem lineages. Each should be given an equal probability for survival unless there is evidence to the contrary.

Lines 120-124. Seilacher’s reason for postulating a mass extinction is well explained but does not apply now since the vendobionts are not outside the Metazoa, according to recent studies. Therefore, it is misleading to say “The case for extinction was strengthened . . .”.

Lines 233+. It is good to see the addition of a section that addresses the null hypothesis, that the extinction of Ediacaran organisms was due to natural attrition not some exotic process. However, the percentage argument for mass extinction is weak, when compared with data from the Big Five during the Phanerozoic. Here, we are considering a trivial number of genera that went extinct when compared with the massive turnovers at the P-T and K-Pg boundaries. A better approach might be to ask how close to the presumed extinction horizon do the taxa in question occur? The unexpected persistence of erniettomorphs to within meters of the Ediacaran-Cambrian boundary in Namibia and Nevada provides support for the mass extinction hypothesis, but we are speaking of only 2-3 taxa, at best. Hence, this reviewer remains unconvinced. The fact that the Nama-aged Shibantan Member (lines 492-494) preserves Avalonian and White Sea taxa only reinforces this scepticism.

---

## [Reviewer Report]

The authors have addressed all my comments and concerns, and in my opinion, done a reasonable job of addressing the specific comments and concerns of the other reviewers. The result is an improved manuscript that I think can be accepted and will be a valuable contribution to ongoing investigation of the Ediacaran-Cambrian boundary.

---

## [Editor Report]

Dear Authors:

Thank you for submitting your revised manuscript and for providing a very detailed explanation of your revisions. Your revised manuscript has been re-reviewed by two of the original reviewers (including the most critical reviewer). The more critical reviewer argues that some revisions are still required and I agree that considering those comments carefully can further improve your manuscript.

Please revise the manuscript to address the suggestions of the reviewer and provide a detailed explanation of your revisions.

I am looking forward to receiving your revised manuscript,

With best regards,

Michal Kowalewski (Handling Editor)

---

## [Editor Report]

Dear Authors,

Thank you for your detailed revisions and detailed responses to comments of the reviewers and the editor. In my opinion all issues have been addressed satisfactorily and I recommend that the paper be accepted for publication in Extinction.

HE